# ExMAG: Learning of Maximally Ancestral Graphs

## Abstract

In mixed graphs, there are both directed and bidirected edges. An extension of acyclicity to this mixed-graph setting is known as maximally ancestral graphs. This extension is of considerable interest in causal learning in the presence of confounders. There, directed edges represent a clear direction of causality, while bidirected edges represent confounding. We propose a branch-and-cut algorithm for learning maximally ancestral graphs. The algorithm produces more accurate results than state-of-the-art methods, while being faster to run on small and medium-sized synthetic instances.

## 1 Introduction

As one transitions from statistical to causal learning Schölkopf & von Kügelgen (2022), one is seeking the most appropriate causal model. Dynamic Bayesian networks (DBN) Dean & Kanazawa (1989); Murphy (2002) are a popular model, where a weighted directed acyclic graph represents causal relationships. The vertices represent stochastic processes, and weighted, oriented edges suggest the strength of the causal relationships. The key challenge in learning DBNs is confounding. A DNB with autoregressive order of $0$ reduces to a Bayesian network (BN) Peal (1985), and posses another challenge during training, because cycles in the causal relationships must be excluded.

To illustrate the challenge of confounding, let us consider Simpson's paradox. Simpson's paradox shows that without considering confounding factors in statistical analysis (McElreath, 2018), the direction of causality can be misestimated completely. A textbook example (McElreath, 2018) comes from the Berkeley graduate admissions Bickel et al. (1977). The data show that women find it harder to get admitted to Berkeley graduate schools. Nevertheless, this is because women tend to apply to departments that have lower admission rates. In this example, the choice of the graduate school is the confounder, impacting the probability of admission. Confounding is prevalent throughout high-dimensional statistics Lin et al. (2014); Gilad & Mizrahi-Man (2015).

Specifically, in biomedical sciences, confounders such as socio-economic status, age, or lifestyle factors can distort the true causal relationship between treatments and outcomes Zhou et al. (2022b). Techniques such as instrumental variables Reiersøl (1945); Imbens (2014), propensity score matching Rosenbaum & Rubin (1983), and double machine learning Chernozhukov et al. (2018) have been widely used to mitigate the effects of confounding in clinical trials and observational studies. To mitigate confounding biases, statistical models that explicitly account for hidden confounders, such as spectral methods and latent variable models, are often employed (Guo et al., 2022). Furthermore, meta-analysis and sensitivity analysis are often used to evaluate the robustness of findings in the presence of potential confounders, especially when combining results from multiple studies Bühlmann (2020); Mathur & VanderWeele (2022).

In statistical theory, work Bühlmann & Geer (2011) studies confounding in detail, and many subsequent works develop further methods. (Lam & Yao, 2012) shows that leveraging the dominant eigenstructure of time series may improve performance of estimation. Anchor regression, for instance, bridges the gap between causality and robustness by addressing heterogeneity in data (Rothenhäusler et al., 2018). Other significant contributions include spectral deconfounding (Bühlmann & Ćevid, 2020). Similarly, the invariance principle has emerged as a cornerstone of causal inference, linking causal structure to robust statistical models (Bühlmann, 2020). Furthermore, the concept of doubly robust inference offers an alternative framework for addressing hidden confounding factors, com-

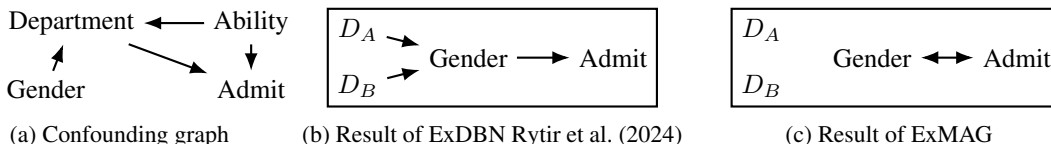

(a) Confounding graph      (b) Result of ExDBN Rytir et al. (2024)      (c) Result of ExMAG

Figure 1: Ground truth with the confounder of Department on the Berkeley graduate admission example (left, 1a), a dynamic Bayesian network trained on the data (center, 1b), and ExMAG output (right, 1c). While the dynamic Bayesian network suggests a causal relationship between gender and admission, ExMAG correctly identifies the confounding with the department serving as a latent confounder. See the supplementary material for details.

bining model robustness with efficiency in high-dimensional scenarios (Guo et al., 2022). Together, these developments represent a significant step forward in understanding and addressing the challenges posed by complex causal systems with missing or latent variables (Wang & Zhou, 2021; Reiser, 2020). While these methods have shown promise, they often rely on simplifying assumptions, such as stationarity or full observability (Bhattacharya et al., 2021).

While there is a long history of the study of confounding, as suggested above, the extensions of DBNs to allow for confounding are rather more recent. Instead of estimating a Directed Acyclic Graph (DAG), one could estimate a Maximal Ancestral Graph (MAG), cf. Richardson & Spirtes (2002). MAGs allow for both direct and indirect relationships among variables modelled as directed and bidirected edges, even in the presence of confounding factors. In particular, MAGs can represent bidirectional relationships that DAG-based models, such as BNs and DBNs, cannot. This makes MAGs a more powerful tool for capturing the complex dynamics of real-world causal systems.

There are only a few studies of MAG estimation Chen et al. (2021); Rantanen et al. (2021); Claassen & Bucur (2022); Hu (2023); Hu & Evans (2024b;a); Dash et al. (2025). (Richardson, 2009; van Ommen, 2024) are applicable to both discrete and nonparametric cases, which extend DAG to MAG or ADMG diagrams. Factorization in MAG is not directly decomposable into individual variables and their parent sets, as in DAGs, but must instead consider components connected by bidirected paths (termed *districts* or *c-components*), cf. (Richardson, 2009), although Claassen & Bucur (2022) proposed to use Markov equivalence classes (MEC) instead. In 2021, Chen et al. (2021) introduced a first mixed-integer programming (MIP) formulation, but the number of variables scales with the number of c-components, i. e., exponentially with the number of vertices in the worst case. Such formulations are also known as extended formulations Conforti et al. (2010). The same year, Rantanen et al. (2021) explored a score-based approach for directed MAG discovery, leveraging a local score function optimized using pruning rules and dynamic programming. Additionally, Zhou et al. (2022a) addresses exogenous covariates in causal formulation that helps explain the heterogeneity in both sampling and causal mechanisms. Hu (2023)'s dissertation presented an extension of the imsets of Studeny (2006) from directed acyclic graphs (DAGs) to towards MAGs Hu & Evans (2024b), which allows for the use of the methods of Studený, and a score-based heuristic Hu & Evans (2024a). More recently, paper Dash et al. (2025) enhanced the scalability of methods of Chen et al. (2021) by utilizing linear programming (LP) relaxations instead of solving the MIP.

Our approach proposes a formulation of MAG estimation within Mixed-Integer Nonlinear Programming (MINLP) in a dimension polynomial in the number of vertices, focusing on latent confounding specifically. From the exponential set of constraints enforcing acyclicity, only those violated by a solution are iteratively added to the program in a lazy manner. In contrast, the so-called extended formulations of Chen et al. (2021); Dash et al. (2025) have the dimension exponential in the number of vertices. While both the extended formulation of Chen et al. (2021) and ours ensure that confounding factors are properly accounted for and the true underlying data-generating process is better represented by the model, our implementation scales further, from 4-5 stochastic processes in the extended formulation of Chen et al. (2021) to 25 or more stochastic processes with the proposed compact formulation.

## 1.1 MOTIVATING EXAMPLE

Let us revisit the Berkeley graduate admission paradox example. As in most paradoxes, there is no violation of logic in Simpson's paradox, just a violation of intuition. In this case, the intuition is that a positive association in the entire population should also hold within each department. Overall, females in these data did have a harder time getting admitted to graduate school. But that arose, because female applicants chose the departments that were the most difficult to gain admission to for anyone, male or female. In this example, gender influences the choice of department, and the department influences the chance of admission. Controlling for department reveals a more plausible direct causal influence of gender, as illustrated in Fig. 1a. Our method, ExMAG, is able to reveal the confounders in this Berkeley graduate admission example, as illustrated in the notebook in the supplementary materials and Fig. 1.

## 2 GRAPHS AND PROPERTIES

A DAG is a directed graph $\mathcal{G} = (\mathcal{V}, \mathcal{E})$ such that there are no directed cycles. That is, there is no sequence of distinct vertices $v_1, v_2, \ldots, v_k \in \mathcal{V}$ such that $(v_i, v_{i+1}) \in \mathcal{E}$ for all $1 \leq i \leq k - 1$ and $(v_k, v_1) \in \mathcal{E}$. Maximal ancestral graphs (MAGs), introduced by Richardson & Spirtes (2002), provide a framework for modeling distributions through conditional independence relations. Compared with directed acyclic graphs (DAGs), MAGs allow for latent confounders, accommodating data that arise from distributions with more complex independence structures and revealing hidden states in the graphs. While DAGs allow for the efficient computation of maximum likelihood estimates (MLEs) and scoring (e.g., via BIC), these properties are challenging to extend to MAG due to their structural and computational complexity (Hu & Evans, 2020).

**ADMG** Mixed graphs feature two types of edges: directed ($\rightarrow$) and bidirected ($\leftrightarrow$). Mixed graph $\mathcal{G}$ thus consists of a vertex set $\mathcal{V}$, a set of directed edges $\mathcal{E}$ and bidirected edges $\mathcal{B}$, where $\mathcal{E}$ are ordered pairs of vertices, while $\mathcal{B}$ are unordered 2-element subsets of distinct vertices. For a directed edge in $\mathcal{E}$ connecting vertices $v$ and $w$, we say these two vertices are the *endpoints* of the edge and the two vertices are *adjacent* (otherwise they are *non-adjacent*). For a vertex $v$ in $\mathcal{V}$, we define the *parents*, *spouses*, and *ancestors* of $v$, respectively as:

$$\mathrm{pa}_{\mathcal{G}}(v) = \{w : w \rightarrow v \text{ in } \mathcal{G}\}, \qquad \mathrm{an}_{\mathcal{G}}(v) = \{w : w \rightarrow \cdots \rightarrow v \text{ in } \mathcal{G} \text{ or } w = v\},$$
$$\mathrm{sp}_{\mathcal{G}}(v) = \{w : w \leftrightarrow v \text{ in } \mathcal{G}\}.$$

As in a DAG, a mixed graph $\mathcal{G}$ is acyclic if it contains no directed cycles in $\mathcal{E}$, i.e., an acyclic directed mixed graph (ADMG) Hu (2023).

A directed mixed graph $\mathcal{G}$ is called an ancestral ADMG if the following condition holds for all pairs of nodes $v$ and $w$ in $\mathcal{G}$:

$$\text{If } v \neq w \text{ and } v \in \mathrm{an}_{\mathcal{G}}(w) \cup \mathrm{sp}_{\mathcal{G}}(w), \text{ then } w \notin \mathrm{an}_{\mathcal{G}}(v),$$

which is written as, $\mathcal{G}$ is an ancestral ADMG if it contains no directed cycles ($v \rightarrow u \rightarrow \ldots \rightarrow w \rightarrow v$) or almost directed cycles Chen et al. (2021); Hu (2023). In an ADMG, an almost directed cycle is of the form $v \rightarrow u \rightarrow \ldots \rightarrow w \leftrightarrow v$; in other words, $\{v, w\} \in \mathcal{U}$ is a bidirected edge, and $v \in \mathrm{an}_{\mathcal{G}}(w)$ Chen et al. (2021).

**Inducing Paths** An inducing path from variable $X$ to variable $Y$ in a directed graph $\mathcal{G} = (\mathcal{V}, \mathcal{E})$ is a path $P = (X = v_0, v_1, \ldots, v_n = Y)$ such that for all intermediate nodes $v_i$ (where $1 \leq i \leq n-1$), $v_i \in Z$, where $Z$ is the conditioning set. If the path is blocked by conditioning on $Z$, then it is an inducing path.

A node $u$ on a non-overlapping path is called a collider if it is contained in an non-overlapping subpath $(w, u, v)$ with two arrowheads into $u$. In mathematical form, a collider is represented as

$$\mathrm{collider}_{\mathcal{G}}(u) = \{u : \exists w, v : \ w \rightarrow u \leftarrow v \vee w \leftrightarrow u \leftarrow v \vee w \rightarrow u \leftrightarrow v \vee w \leftrightarrow u \leftrightarrow v \ \}.$$

**m-separation** Graphs encode conditional independence via separation criteria. For acyclic directed mixed graphs (ADMGs), *m-separation* generalizes d-separation to handle bidirected edges. A path between nodes $a$ and $b$ is *m-connecting* given a conditioning set $C \subseteq \mathcal{V}$ if: (i) $a$ and $b$ are the endpoints; (ii) all non-colliders are not in $C$; and (iii) all colliders are in $\mathrm{an}_G(C)$. Nodes $a$ and $b$ are *m-separated* given $C$ if no such path exists.

**Maximal Ancestral Graph**   An ADMG $\mathcal{G}$ is called a maximal ancestral graph (MAG) if:

(i) For every pair of nonadjacent vertices $a$ and $b$, there exists some set $C$ such that $a, b$ are *m-separated* given $C$ in $\mathcal{G}$ (*Maximality*);

(ii) For every $v \in \mathcal{V}$, $\mathrm{sib}_{\mathcal{G}}(v) \cap \mathrm{anc}_{\mathcal{G}}(v) = \emptyset$ (*Ancestrality*).

where $\mathrm{sib}_{\mathcal{G}}(v) = \{u \in \mathcal{V} \mid \exists w \in \mathcal{V} : w \to v, \ w \to u, \ u \not\to v, \ v \not\to u\}$. We refer to Hu (2023) for multiple examples.

## 3   STRUCTURAL EQUATION MODEL WITH LATENT CONFOUNDERS

Recent works on high-dimensional confounding or deconfounding clarify the connections between distributional robustness, replicability, and causal inference (Rothenhäusler et al., 2018; Guo et al., 2022). Distributional robustness differs significantly from traditional robust statistical methods Huber (1964); Hampel et al. (1986), which typically handle outliers in the training data, while our work focuses on evaluating the existence of a confounding factor.

In this section, we inherit from distributional robustness and present the formulation of the underlying Structural Equation Model (SEM). Since we cannot observe all relevant variables, we must deal with the situation of hidden confounding. The problem is formalized in the following form corresponding to a SEM Bollen (1989); Pearl (2009):

$$X \leftarrow XW_D + LW^L + \epsilon,$$

where $X = (X_1, \ldots, X_d)$ is the vector of variables, $L = (L_1, \ldots, Lp)$ is the vector of latent variables, $W_D$ is a $\mathbb{R}^{d \times d}$ weighted adjacency matrix of the DAG encoding the causal structure, $W^L \in \mathbb{R}^{d \times p}$ encodes structure of latent variables, and $\epsilon$ is the noise vector.

Further, we will assume that latent confounding term $LW^L$ has been absorbed into the error term $\epsilon$.

$$X \leftarrow XW_D + \epsilon.$$

The covariance matrix $B = \mathbb{E}[\epsilon \epsilon^T]$ will represent the hidden confounders.

If each $\epsilon_i$ is Gaussian, then the resulting distribution $p(X)$ is multivariate normal with zero mean and covariance

$$\Sigma = (I - W_D)^{-T} W_B (I - W_D)^{-1}.$$

The cost function then becomes the error of the prediction, represented by the difference between the sampled data $X$ and their prediction stemming from observed variables, and influece of latent confounders, similarly to Bhattacharya et al. (2021).

$$J(W_D, W_B) = \|X - XW_D - (X - XW_D)W_B\|^2$$

### 3.1   CONNECTION TO CAUSALITY

The causal parameter $W_D$ and $W_B$ can be seen as minimizing the worst-case risk:

$$\arg \min_{W_D, W_B} \max_{P \in \mathcal{P}} \mathbb{E}_P \left[ (X - XW_D - (X - XW_D)W_B)^2 \right],$$

where $\mathcal{P}$ is a class of distributions containing perturbations of the original distribution, including confoundings. Class $\mathcal{P}$, therefore, guides the overall structure of the MIQP formulation. This modeling highlights the inherent connection between causality and distributional robustness (Dawid & Didelez, 2010; Peters et al., 2016; Rojas-Carulla et al., 2018).

Now we denote by $X$ the data matrix. Let $X \in \mathbb{R}^{n \times d}$ be a $n \times d$ matrix with data samples. Then under assumption of Gaussian noise, the problem can be reformulated as

$$\arg \min_{W_B, W_D} J(W_D, W_B) \tag{1}$$

# 4 FORMULATION OF THE MIXED INTEGER QUADRATIC PROGRAM

Here, we present the formulation of the Mixed Integer Quadratic Program (MIQP) used to infer the causal structure, with a new binary matrix $B = [b_{j,k}] \in \{0,1\}^{d \times d}$ introduced to account for relationships explained by confounding factors, alongside the weight matrix $W = [w_{j,k}] \in \mathbb{R}^{d \times d}$ and binary adjacency matrix $E = [e_{j,k}] \in \{0,1\}^{d \times d}$ adopted from the ExDAG model Rytíř et al. (2024). Whenever entry $w_{j,k}$ is non-zero, either $e_{j,k}$ (directed edge) or $b_{j,k}$ is nonzero (bidirected edge). At the same time, we extend the existing formulation by introducing an additional binary input matrix $F = [f_{j,k}] \in \{0,1\}^{d \times d}$, where $f_{j,k}$, indicating that there is no direct causal relationship between variables $j$ and $k$, but $j$ and $k$ might have a common cofactor. This follows from the meaning of the edges in a MAG $\rightarrow$ edge implies a direct causal relationship but does not rule out a possible latent confounding, $\leftrightarrow$ means no direct causal relationship.

Formally, we define *Directed Edge Matrix* $E$ as $e_{j,k} = 1$ if $j \rightarrow k$, 0 otherwise. Similarly, *Bidirected Edge Matrix* $B$ is $b_{j,k} = 1$ if $j \leftrightarrow k$, and 0 otherwise. Lastly, the input matrix $F$ with pairs of variables which are known not to be in a direct cause-effect relationship, is by definition $f_{j,k} = 1$ if $j \nrightarrow k \wedge k \nrightarrow j$ and 0 otherwise. Note that this matrix is by definition symmetric.

## 4.1 MILP FORMULATION

The cost function for the Mixed Integer Quadratic Program of ExMAG is the $l_q$ norm below. It has two components - the error of prediction of $X_{i,j}$, and the regularization term. Denote

$$R_{i,j} = X_{i,j} - \sum_{k=0; k \neq j}^{d} X_{i,k} w_{Dk,j}.$$

Then, the minimization problem in equation 1 corresponds to the minimization of the following cost function

$$\min_{W_D, W_B, E, B} \sum_{i=1}^{n} \sum_{j=1}^{d} \left| R_{i,j} - \sum_{k=0; k \neq j}^{d} R_{i,k} w_{Bk,j} \right|^q + \lambda \sum_{j=0}^{d} \sum_{k=0}^{d} (e_{j,k} + b_{j,k}), \quad (2)$$

In this formula, $X_{i,j}$ represents the value of the $j$-th variable for the $i$-th data point; $w_{Dk,j}$ represents the weight of the directed edge from variable $k$ to variable $j$; $w_{Bk,j}$ represents the weight of the bidirected edge from variable $k$ to variable $j$; $e_{j,k}$ is the binary decision variable indicating the presence of a directed edge from $j$ to $k$; $b_{j,k}$ is the binary decision variable indicating a bidirected edge between $j$ and $k$; $\lambda \in \mathbb{R}^{+}$ is a regularization parameter controlling the model fit and the edge penalty trade-off. The exponent $q \in \mathbb{N}$ can take values $q = 1$ or $q = 2$.

Optimization criterion in equation 2 implies that the dependencies between the variables are linear. The first part of the criterion encodes for the actual cost as an error of the prediction, the second part encodes for regularization, penalizing more edges with a larger $\lambda$.

As in the ExDAG Rytíř et al. (2024) model, the weights are bounded by introducing a large constant $c$, which is chosen large enough to exceed the maximum weight expected in the problem being solved. The bounding avoids bilinear terms in the cost function in equation 2 and takes the following form:

$$-c \cdot E \leq W_D \leq c \cdot E \qquad \text{(Weight Constraint)}$$
$$-c \cdot B \leq W_B \leq c \cdot B \qquad (3)$$
$$E + B \leq \mathbf{1} \qquad \text{(Edge Constraint)}$$

The Edge Constraint means that there cannot be a directed as well as a bidirected edge between the same two vertices. Additionally, we enforce that the bidirected matrix is symmetric by equation 4. If $f_{j,k} = 1$, then we know there is no direct causal relationship between $j$ and $k$, and therefore, $e_{j,k} = 0$. This is formally enforced by equation 5 Inversely, $f_{j,k} = 0$ implies a directed edge rather than a bidirected edge between $j$ and $k$ in equation 6.

$$B = B^T, \qquad (4)$$
$$F + E \leq \mathbf{1}, \qquad (5)$$
$$B \leq F. \qquad (6)$$

Lastly, we must enforce conditions for directed or almost directed cycles and inducing paths. Those conditions are enforced lazily using a separation routine explained later. Directed cycles are enforced in a way adopted from Rytíř et al. (2024). Therefore, they are left out of this paper. An almost directed cycle formed by edges in set $E'$ and a bidirected edge $(u, v)$ is forbidden by the constraint

$$b_{u,v} + \sum_{(j,k) \in E'} e_{j,k} \leq |E'|. \qquad \text{(Acyclic Constraint)}$$

Similarly, if there is an inducing path formed by path $P$ that contains bidirected edges, and set $E'$ contains all directed edges that participate in the ancestor relationship (including multiple paths) between the inner points of the path and the terminals of $P$, this inducing path is forbidden by

$$\sum_{(j,k) \in P} b_{j,k} + \sum_{(j,k) \in E'} e_{j,k} \leq |E'| + |P| - 1. \qquad \text{(Inducing-Paths Constraint)}$$

Note that the second condition does not necessarily eliminate the inducing path, as the optimizer might forbid one of the edges in $E'$ without influencing the ancestor relationship. This results in path $P$ being found in the next iteration, with a smaller set of directed edges, and the process is repeated.

By enforcing these constraints, we ensure that the MIQP correctly models the causal relationships between the variables while respecting the structure defined by $f_{j,k}$ and the potential confounding relationships captured by $b_{j,k}$.

## 5 Separation Routine for the Maximal Ancestral Graphs

The main contribution of this section is the separation routine that identifies whenever a graph is an instance of a maximal ancestral graph. To do so, we need to identify directed cycles, almost directed cycles, and inducing paths. The presence of directed cycles can be detected in $\mathcal{O}(d^2)$ using depth-first-search (DFS); such an approach can be found in Rytíř et al. (2024). For both inducing paths and almost directed cycles, we will use the distance matrix $D$ constructed on the graph of directed edges $E$. This distance matrix can be obtained, for example, using the Floyd-Warshall algorithm Floyd (1962).

Having the distance matrix, to check for almost directed cycles, we can iterate over all bidirected edges and test whether the distance between the endpoints using $E$ is finite, i.e., we have a directed path connected by a bidirected edge. See Algorithm 1 for details.

---
**Algorithm 1** Function that identifies almost directed cycles.
___
**Input:** directed edges $E$, bidirected edges $B$

  **function** Almost-Directed-Cycles($E$, $B$)
      $D \leftarrow$ Distance-Matrix($E$)
      **for all** $(j, k) \in \{1, 2, \ldots, d\} \times \{1, 2, \ldots, d\}$ **do**
        **if** $j \neq k$ & $b_{j,k} == 1$ & $D_{j,k} < \infty$ **then**
          $E' =$ Trace-Distance-Matrix($D,E,j,k$)     ▷ Finds all edges on any $j$ to $k$ path, see Supl.
          Found cycle formed by edges $E'$ and $j \leftrightarrow k$

---

In the case of inducing paths, we use a DFS starting from each vertex. Once started from vertex $s$, the DFS routine checks for all possible inducing paths that terminate in $s$. For efficiency, a set of all possible endpoints of the path is held. Once this set is empty, the DFS search is terminated, and no further exploration is performed. The set is updated using the distance matrix calculated on the directed edges. If we consider a vertex $v$, $s$ must be either its ancestor (meaning that possible endpoints for $v$ remain unchanged) or the second inducing path endpoint is among the points that are reachable from $v$ (meaning that the possible endpoints for $v$ are replaced with their intersection with the set of all points reachable from $v$). See Algorithm 2 for details.

Once having the bidirected edges in the inducing paths and almost directed cycles, we need to trace back the Floyd-Warshall distance matrix to find all directed edges that form the cycle or the ancestor

---

**Algorithm 2** Function that identifies inducing paths.

---

**Input:** directed edges $E$, bidirected edges $B$

   **function** INDUCING-PATHS($E$, $B$)

      $D \leftarrow$ DISTANCE-MATRIX($E$)

      **for all** $s \in 1, 2, \ldots, d$ **do** INDUCING-PATHS-DFS($D$, $E$, $B$, $s$, $s$, $\{1, 2, \ldots d\}$, $[s]$)

   **function** INDUCING-PATHS-DFS($D$, $E$, $B$, $s$, $u$, possible endpoints, path)

      **if** possible endpoints are empty **then return**

      **if** LEN(path) $> 2$ & $u$ in possible endpoints **then** FOUND-INDUCING-PATH($D$,$E$, path)

                        ▷ Edges participating in ancestor relation are recovered, see Suppl. mat.

      **for all** $v \in 1, 2, \ldots, d$ such that $e_{u,v} = 1$ **do**

         v-endpoints $\leftarrow$ possible endpoints **if** $D_{v,s} < \infty$ **else** possible endpoints $\cap \{x \mid D_{v,x} < \infty\}$

         INDUCING-PATHS-DFS($D$,$E$,$B$,$s$,$v$,v-endpoints,path+$v$)

---

relationship. This is done using calls to the function TRACE-DISTANCE-MATRIX, which can be found in the Supplementary materials.

If directed cycles, almost directed cycles, and inducing paths are found, the algorithm applies lazy constraints in Acyclic Constraint and in Inducing-Paths Constraint. Note that removing one directed edge between two vertices where multiple paths exist is not a necessary condition for the graph to become a MAG; however, this procedure can be repeated iteratively. If no inducing paths or almost directed cycles are found, we know that the program converged to the optimum, and we have a maximal ancestral graph, which minimizes equation 2.

## 6 EXPERIMENTAL EVALUATION

**Used Datasets** We tested the ExMAG algorithm on both synthetic and real-world datasets. The first synthetic data set is based on the *Erdös-Rényi model* (ER) Erdős & Rényi (1959), in which the ground truth graph is randomly selected from all graphs with $d$ vertices and $m$ edges (parameter of the experiment, for example, dataset ER-2 contains 2 edges per variable, that is, $m = 2 \cdot d$). The weights of the graph are randomly sampled from the set $(-2.0, -0.5) \cup (0.5, 2.0)$.

Once the ground truth model is created, the training data are generated using the structural model equation. Then, $20\%$ of variables are treated as latent variables and hidden from the training data. The respective columns and rows from the ground truth weight matrix $W$ have also been removed. Finally, $20\%$ of edges between variables not connected by an edge in the ground truth data are marked in $F$.

The second dataset uses randomly generated *bow-free* (BF) graphs. A bow-free graph is a graph such that for no pair of vertices $i, j$, $i \rightarrow j$ and at the same time $i \leftrightarrow j$. The BF graph generation process has two parameters: the probability of a directed edge and the probability of a bidirected edge. The generation process is as follows. First, a bow-free graph with the given edge probabilities is generated randomly. Then the weights of the sampled graphs are randomly sampled from the set $(-2.0, -0.5) \cup (0.5, 2.0)$.

The third synthetic dataset, 3BF, is a modified version of BF. We generate the ground truth graph in the same way as in the BF dataset and then modify it. Specifically, we identify each vertex with a degree greater than three, and randomly remove edges until the vertex has a degree of at most three.

The adjacency matrix of the directed edges defines the weights of the structural equation model. Then the data samples are generated using the structural equation, where the noise is sampled from a multivariate Gaussian distribution with a covariance matrix equal to the adjacency matrix of bidirected edges generated in the previous step.

The fourth dataset uses real-world data from the *financial* sector. Paper Ballester et al. (2023) works with systemic credit risk, one of the most important concerns within the financial system, using dynamic Bayesian networks. The data show that transport and manufacturing companies are likely to transfer risk to other sectors, while banks and the energy sector are likely to be influenced by the risks from other sectors. The data from Ballester et al. (2023) contains a 10-time series capturing

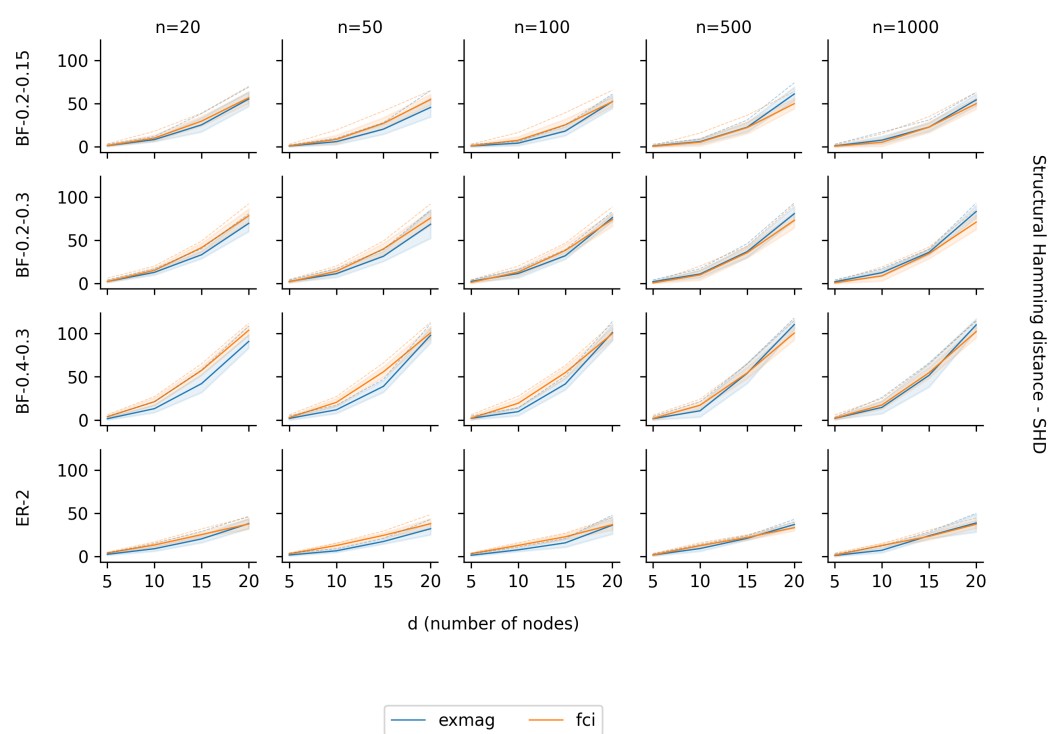

Figure 2: SHD values (in the vertical axis) for different settings of $d$ (in the horizontal axis) and $n$ (horizontal choice of the graph). The plots in the vertical dimension differ according to the dataset used. Standard deviations are depicted as the blurred regions, and dashed lines are the maximum values. See supplementary materials for results on more datasets and error information.

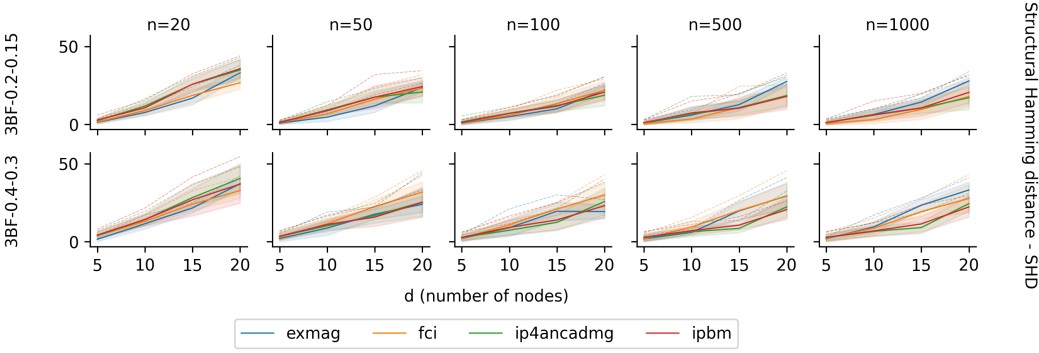

Figure 3: SHD values on 3BF datasets.

the spreads of 10 European credit default swaps (CDS), and further six time series are added from Rytíř et al. (2024).

We set matrix $F$ to encode for no direct causal relationship between any two pairs of companies from different sectors. Banks sector includes 48DGFE, 05ABBF, 8B69AP, 06DABK, EFAGG9, 2H6677, FH49GG, and 8D8575. Insurance sector includes GG6EBT, DD359M, and FF667M. And lastly, transportation sector and manufacturing includes 0H99B7, 2H66B7, 8A87AG, NN2A8G, and 6A516F.

**Evaluation Criteria** Suppose that a tested algorithm produced weight matrix $\hat{W}$. Such a matrix can contain nearly zero weights. For such reasons, thresholding is done, keeping only edges with a

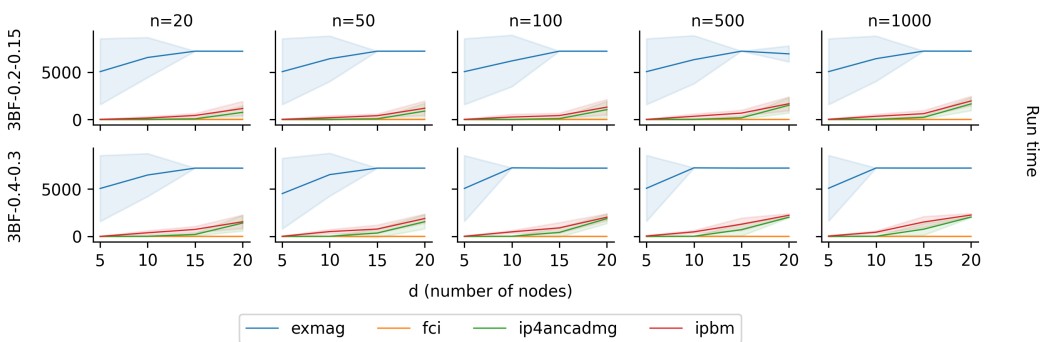

Figure 4: Run times of the compared algorithms in seconds.

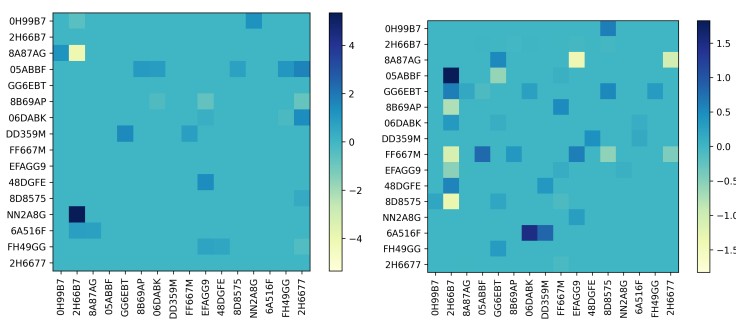

Figure 5: Heatmap of weight matrix $W$ (left) and bidirectional weight matrix $W$ (right) on the financial dataset.

weight greater than or equal to $\delta$. In cases when the ground-truth weight matrix $W$ is known, the best solution (in terms of structural Hamming distance, see below) is kept over those defined by different threshold $\delta$ values. In the evaluation, we use the *structural Hamming distance (SHD)*. This distance is the sum of contributions over all pairs of variables in the graph. For two variables $i, j$, let $GT \in \{\rightarrow, \leftarrow, \leftrightarrow, \emptyset\}$ be the edge type in the ground truth graph and $PR \in \{\rightarrow, \leftarrow, \leftrightarrow, \emptyset\}$ be edge type in the predicted graph. Then the contribution of $i, j$ pair to SHD is $r_{ij} = 0$ if $GT = PR$, $0.5$ if $GT \neq PR \land GT \neq \emptyset \land PR \neq \emptyset$, and $1$ otherwise. Other measured criteria include *runtime* and *F1-score*, i.e., the harmonic mean of precision and recall.

**Experiment Setting**   In the experiments, we show the results of ExMAG. In the case of synthetic datasets, we generated random graphs with the number of vertices $d \in \{5, 10, 15, 20, 25\}$. The number of samples was $n \in \{20, 50, 100, 500, 1000\}$, and for the ground-truth graph, the edge-to-vertex ratio was in $\{2, 3, 4, 5, 6, 7, 8, 9, 10, 15, 20\}$. All tested algorithms were run 10 times, each time on synthetic data generated using a random generator initialized with a different seed. The results were then averaged. We compared our method with the FCI algorithm Spirtes et al. (1995), ip4ancadmg Chen et al. (2021), and ipbm Dash et al. (2025). We set regularization coefficient $\lambda$ to $1.0$. We ran experiments on a computing cluster with AMD EPYC 7543 cpus and each job had allocated two cores and 64GB RAM. Time limit was 900 seconds for ExMAG and 1800 seconds for other methods. The total cpu time needed for experiments in this paper was around one month.

**Experimental Results**   The SHD results are shown in Figures 2 and 3 and in the supplementary materials for additional datasets. The plots show a comparison of SHD values for ExMAG on the synthetic datasets. As can be seen, the structural Hamming distance grows with the number of variables and decreases with the number of samples.

As we can see on Figures 2 and 3, ExMAG performs better than FCI on all scenarios. Since both ipbm and ip4ancadmg have a preprocessing step that depends exponentially on the maximum in-degree of the underlying ground truth graph, we tested these two algorithms only on the 3BF

datasets, where the in-degree is bounded by three. We can see in Figure 3, that ExMAG also performs better than ipbm and ip4ancadmg. The run times of the evaluated algorithms are shown in Figure 4. For additional results (incl. the F1-score), please see the supplementary materials.

The results on the real-world dataset can be seen in Figure 5. Contrary to the original expectations, the highest risk importer is company 2H66B7, which stands for Lufthansa. The second highest risk importer is 2H6677, i.e., the Deutsche Bank, which is an expected result.

## 7 CONCLUSION AND LIMITATIONS

Learning of Bayesian networks has received considerable attention as a means of causal learning. With a few exceptions, the research has not considered confounding explicitly. Our method, Ex-MAG, estimates a maximally ancestral graph, capturing confounding and causal relationships using bidirected and directed edges in a mixed graph. The method provides state-of-the-art statistical performance.

As with many other methods for causal learning, the scalability of the method may leave space for improvement. Although the branch-and-cut algorithm runs in time that is exponential in the number of time series in the worst case, Figure 4 illustrates that our run time is lower than those of ip4ancadmg Chen et al. (2021) and ipbm Dash et al. (2025), while improving the SHD of both recent competitors at the same time (cf. Figure 3). One could improve upon the run-time further by introducing additional cutting planes and more elaborate data structures for the separation of Acyclic Constraint and Inducing-Paths Constraint, perhaps drawing inspiration from solvers (Cook et al., 2011, e.g.) for the travelling salesman problem.

In terms of future work, exploring the predictive power of forecasting using variants of dynamical Bayesian networks with confounding considerations seems prominent. Although it seems clear that marginalization is hard even in dynamical Bayesian networks, and thus the computational complexity may be high, but statistical performance is likely to improve, when confounding is considered.

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

# SUPPLEMENTARY MATERIALS

## A    F1-SCORE RESULTS

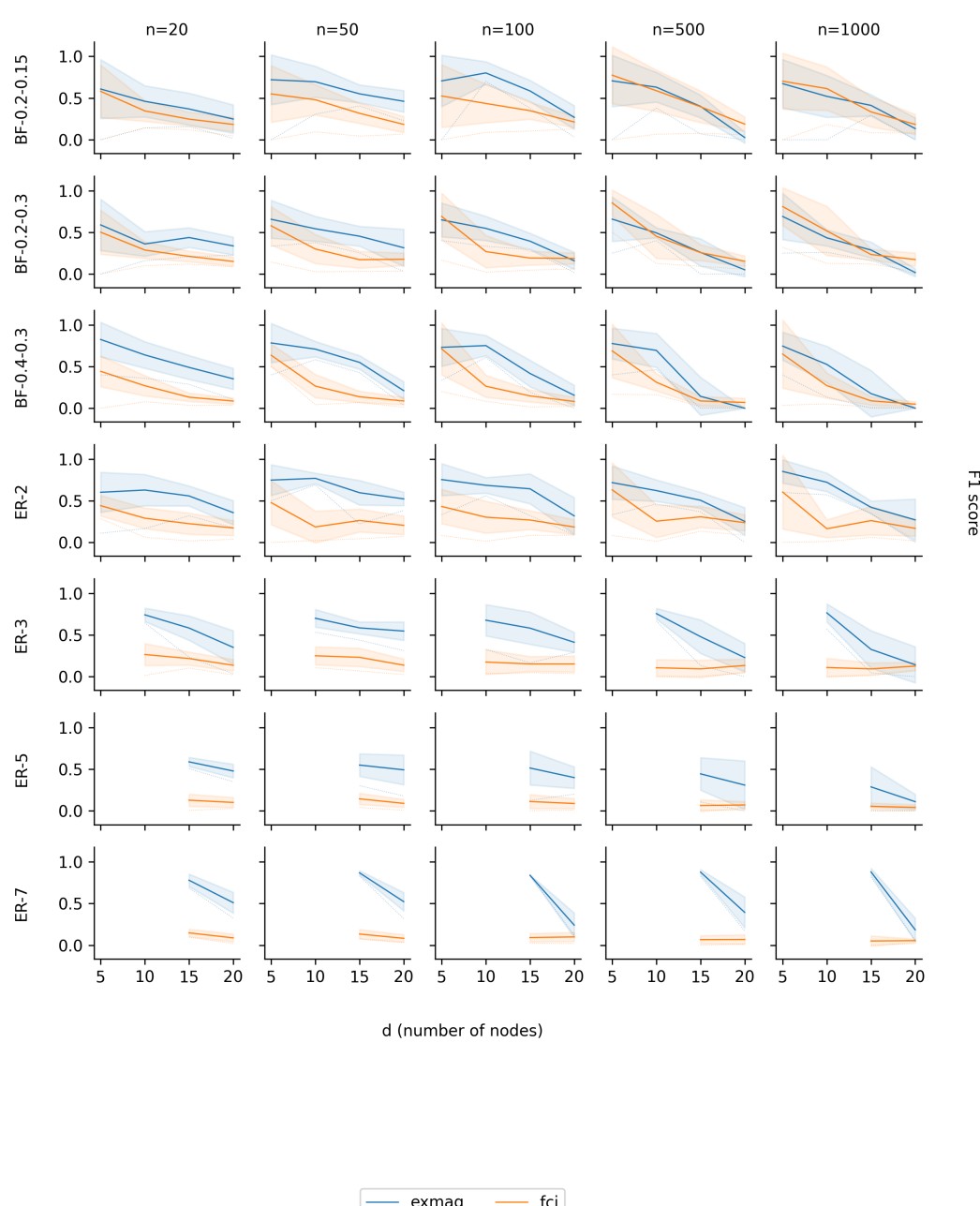

Figure 6: F1-score (in the vertical axis) for different settings of $d$ (in the horizontal axis) and $n$ (horizontal choice of the graph). The plots in the vertical dimension differ according to the dataset used. Standard deviations are depicted as the blurred regions, and dashed lines are the minimum values. Please note that for some of the ER plots, the graphs can be generated only for higher numbers of variables. For example, there exists no ER-5 with $d = 10$, as it would need to contain 50 edges, while the maximum is 45.

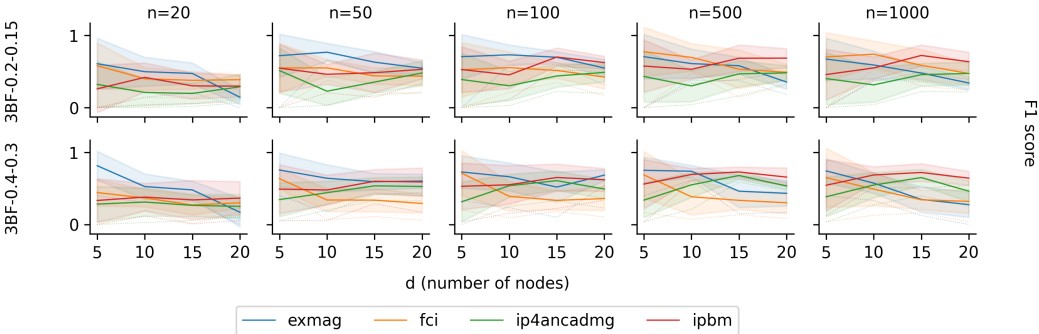

Figure 7: F1-score comparison of all evaluated algorithms on 3BF datasets.

## B SHD RESULTS

Figure 8: SHD values (in the vertical axis) for different settings of $d$ (in the horizontal axis) and $n$ (horizontal choice of the graph). The plots in the vertical dimension differ according to the dataset used. Standard deviations are depicted as the blured regions and dashed lines are the maximum values.

## C  TABLE OF NOTATION

Table 1: A Table of Notation.

| Symbol | Representation |
| --- | --- |
| $G$ | Gender |
| $D$ | Department |
| $A$ | Admit acceptance |
| $U$ | potential confounder, is academic ability in this example |
| $\mathcal{G}$ | Graph |
| $\mathcal{V}$ | Vertex set of the graph |
| $\mathcal{E}$ | Set of directed edges |
| $\mathcal{B}$ | Set of bidirected edges |
| $\mathrm{pa}_{\mathcal{G}}(v)$ | Parents of vertex $v$ in graph $\mathcal{G}$ |
| $\mathrm{sp}_{\mathcal{G}}(v)$ | Spouses of vertex $v$ in graph $\mathcal{G}$ |
| $\mathrm{an}_{\mathcal{G}}(v)$ | Ancestors of vertex $v$ in graph $\mathcal{G}$ |
| $\mathrm{dis}_{\mathcal{G}}(v)$ | District of vertex $v$ in graph $\mathcal{G}$ |
| $\mathrm{collider}_{\mathcal{G}}(u)$ | Collider nodes in graph $\mathcal{G}$ |
| $W_D$ | Directed eight matrix |
| $W_B$ | Bidirected eight matrix |
| $E$ | Directed edge matrix |
| $B$ | Bidirected edge matrix |
| $F$ | Idicates that two variables have no direct causal relationship |
| $X_{i,j}$ | Value of the $j$-th variable for the $i$-th data point |
| $w_{Dk,j}$ | Weight of the directed edge from variable $k$ to variable $j$ |
| $w_{Bk,j}$ | Weight of the bidirected edge from variable $k$ to variable $j$ |
| $e_{j,k}$ | Binary variable indicating a directed edge from $j$ to $k$ |
| $b_{j,k}$ | Binary variable indicating a bidirected edge between $j$ and $k$ |
| $f_{j,k}$ | Binary variable indicating a existence of directed edge between $j$ and $k$ |
| $d_{j,k}$ | Binary variable indicating a directed edge between $j$ and $k$ |
| $r_{ij}$ | Contribution of pair $(i, j)$ to SHD |
| $\lambda$ | Regularization parameter |
| $c$ | Large constant for weight bounding |
| $d$ | Number of variables |
| $n$ | Number of data points |
| $q$ | Exponent in the cost function ($q = 1$ or $q = 2$) |
| $GT$ | Edge type in the ground truth graph |
| $PR$ | Edge type in the predicted graph |
| $\delta$ | Threshold for edge weights |
| $X$ | Causal variable |
| $\epsilon$ | Noise term |
| $\mathcal{P}$ | Class of distributions |
| $C$ | Conditioning set |

## D  PSEUDOCODE

---

**Algorithm 3** Functions that help in the separation routine.

---

**Input:** distances $D$ defined by directed edges $E$, start point $j$, and endpoint $k$
**Output:** edges on any path from $j$ to $k$

    **function** TRACE-DISTANCE-MATRIX($D, E, j, k$)
        **if** $D_{j,k} == \infty$ **then return**
        visited = $\{(j, k)\}$
        stack = stack with $(j, k)$
        edges = $\{\}$
        **while** stack is not empty **do**
            $u, v \leftarrow$ POP(stack)
            **for all** $w \in 1, 2, \ldots, d$ s t. $D_{u,w} + D_{w,v} < \infty$ **do**
                visited $\leftarrow$ visited $\cup \{(u, w), (w, v)\}$
                **if** $E_{u,w}$ **then** edges $\leftarrow$ edges $\cup \{(u, w)\}$
                **else if** $E_{w,v}$ **then** edges $\leftarrow$ edges $\cup \{(w, v)\}$
                add $\{(u, w), (w, v)\}$ to stack
        **return** edges

    **function** FOUND-INDUCING-PATH($D, E, P$)
        $E' = \{\}$
        **for all** vertices $j \in P$ and $j \notin \{P_0, P_{|P|}\}$ **do**
            $E' = E' \cup$ TRACE-DISTANCE-MATRIX($D, E, j, P_0$) $\cup$ TRACE-DISTANCE-MATRIX($D, E, j, P_{|P|}$)
                                             $\triangleright$ Finds all edges on any $j$ to $P_0$ ($P_{|P|}$) path
        Found inducing path formed by path $P$ and directed edges $E'$

---

## E  RESIDUAL ASSUMPTIONS AND STATISTICAL SIGNIFICANCE

There exists a fundamental premise in structural equation modeling that the residuals—representing unexplained variation—are asymptotically unbiased, meaning they are independent of both observed and latent variables, and follow a zero-mean distribution. This assumption plays a critical role in ensuring that learned causal relationships are not distorted by hidden confounders or systematic error. The ExMAG framework embraces this principle by design, explicitly modeling residual independence as a safeguard against spurious causal edges. Just as fairness-aware systems aim to isolate structural patterns from social bias Barocas et al. (2019), ExMAG works to separate signal from statistical noise. The result is a model capable of learning causal mechanisms that are not only mathematically sound but also resilient across different subpopulations, forming a foundation of causal inference.

Causal discovery systems, like decision-making algorithms in high-stakes domains, must operate effectively across structurally diverse populations. This paper uses real-world financial data spanning multiple sectors—banking, insurance, manufacturing, and transportation—each exhibiting distinct systemic exposures. These domains can be viewed as a *privileged setting* where data availability and quality are high, yet subgroup heterogeneity remains significant. In such contexts, ExMAG successfully identifies dominant risk propagation patterns, even when feature distributions vary across industries. This mirrors broader challenges in fairness: the need to perform robustly across populations with unequal baseline conditions Mehrabi et al. (2021). The model's consistent recovery of risk links—illustrated in Figure 5—not only affirms its structural fidelity but also its capacity to generalize without group-specific tuning.

Understanding the statistical reliability of a model's output requires more than average performance—it demands insight into variance. To that end, the authors conduct 10 independent trials for each configuration, reporting both mean and standard deviation for key metrics such as SHD and F1-score. The inclusion of error bars in Figures 2 and 6 provides a visual representation of variability, revealing not just how well the model performs, but how consistently. In contrast to baseline methods with large fluctuations, ExMAG demonstrates narrow error margins, underscoring its stability in the face of stochastic elements like data partitioning and initialization.

## F  BRIEF INTRODUCTION TO MIXED INTEGER QUADRATIC PROGRAMMING

Let us also provide a short introduction to mixed-integer quadratic programming. An optimization problem is called a mixed-integer quadratically constrained quadratic program (MIQCQP) if it is of the form

$$\min_{x \in \mathbb{R}^n} \quad x^T Q x + q^T x, \tag{7}$$

$$\text{s.t. } x^T Q_i x + q_i^T x \le a_i, \tag{8}$$

$$Ax \le b, \tag{9}$$

$$x \in \mathbb{R}^{n-r} \times \mathbb{Z}^r \tag{10}$$

where $Q, Q_i \in \mathbb{R}^{n \times n}$, $q, q_i \in \mathbb{R}^n$, $A \in \mathbb{R}^{m \times n}$, $a \in \mathbb{R}^k$, $b \in \mathbb{R}^m$ and $m, n, k, r \in \mathbb{N}$. Equation 7 is called the cost or loss function, equation 8 represents the quadratic constraints, equation 9 are the linear constraints, and equation 10 enforces the integrality constraints for the last $r$ components of the vector of decision variables $x$.

Mixed-integer quadratic programs have been shown to be NP-hard Del Pia et al. (2014), which often leads to an exhaustive demand for computational resources. The algorithms used to solve MIQP are typically branch-and-bound or cutting plane Dakin (1965); Bonami et al. (2011); Westerlund & Pettersson (1995); Kronqvist et al. (2015). Both of these algorithmic treatments are often employed together, often with the addition of a presolving step, the use of heuristics, and parallelism. The aforementioned allows many modern solvers to solve even large problems despite the NP-hardness. Some of these solvers are open source (like SCIP and GLPK), and others are commercial (GUROBI and CPLEX). The powerful infrastructure present in these solvers can be made use of together with additional problem-specific modifications to deliver high-quality solutions.

Due to the exhaustive nature of the algorithms mentioned in the previous paragraph, global convergence is guaranteed Belotti et al. (2013). Furthermore, convergence to the global solution may be tracked and the error estimated by computing the dual problem of (7–10). The dual of the problem is then used to compute the so-called MIP GAP as follows

$$\text{MIP GAP} = \frac{|J(x^*) - J_{\text{dual}}(y^*)|}{|J(x^*)|}, \tag{11}$$

where $x^*$ and $y^*$ are the current best solutions of the primal and dual problems, respectively, and $J$ and $J^*$ are the cost functions of the primal and dual problems, respectively. The MIP GAP ensures that we can assess the quality of the minimization during solution time and terminate the computation when the result is good enough (small enough MIP GAP). Furthermore, if the gap reaches 0 at any point, we are sure that the current solution is a global optimum.

## G  USE OF LLMS

During writing this paper, Large Language Models were used in the following ways:

- for fixing typos in the paper and to search for alternative formulations (e.g., Grammarly, Writeful, ChatGPT and similar);
- to search for related works (e.g., ChatGPT, Gemini used in Google search and similar);
- for generating straightforward parts of the code, especially data manipulation (IDE integrated AI-based code completion, ChatGPT and similar).

The LLMs contributed below contributing level author, LLMs, however, helped the authors with straightforward and repetitive tasks.

