# OpenReview forum: "ExMAG: Learning of Maximally Ancestral Graphs"
_ICLR.cc/2026/Conference — ICLR 2026 Conference Desk Rejected Submission_

### Official Review · Reviewer_1v33 · 2025-10-24

**Soundness:** 1
**Presentation:** 1
**Contribution:** 1
**Rating:** 0
**Confidence:** 5

**Summary:**

The paper proposes a score-based causal discovery algorithm for learning maximal ancestral graphs (MAGs) based on mixed integer quadratic programming, which relies on a penalized least squares loss assuming linear dependencies. To ensure that the learned graph is a MAG, separation routines are used that detect (almost) directed cycles and inducing paths. It is claimed that the proposed methods yield favorable results on synthetic data sets compared to other methods.

**Strengths:**

Score-based causal discovery is an important topic and in particular for learning of MAGs it is underexplored (even though exact and heuristic algorithms exist). The paper tackles the problem with quadratic programming by assuming linear dependencies and working with a least squares based loss function, which is a novel idea in this specific setting.

**Weaknesses:**

The core weaknesses of the paper are its missing rigor and the overall poor presentation.

**Missing rigor in the score function:**
As far as I can tell, the score function (4) is proposed ad hoc and without any clear derivation or reason why the graph optimizing this function should (asymptotically) coincide with the underlying causal structure. Even more, the proposed score is based on the least squares loss, which in the context of DAG learning has been observed to be consistent only under an equal variance assumption and is generally susceptible to variable scaling. The paper should also emphasize earlier in the paper that linear dependencies are assumed (unless I missed something, this is not mentioned before page 5).

**Presentation:**
The presentation of the paper is lacking in many regards. Apart from typos, there are questionable statements, which at the very least confuse the reader:
- Directly in the abstract: "In mixed graphs, there are both directed and undirected edges. An extensions [...] to this setting is known as maximally ancestral graphs. [...] the undirected edges represent confounding.", only later it is clarified that the undirected edges refer to bidirected edges
- In the first paragraph "Dynamic Bayesian networks (DBN) are a popular model where a DAG represents causal relationships. The key challenge in learning DBNs is confounding.", with large parts of the paper then talking about DAGs and MAGs without the time series setting that DBNs address.
- On the second page "MAGs can represent feedback loops", which is simply a false statement.

The introduction to MAGs in section 2 lacks rigor and is also hard-to-follow for readers not already familiar with the topic. As an example consider the treatment of inducing paths, which are (i) defined incorrectly with regard to some conditioning set Z (ii) based on being "blocked by conditioning on Z", a notion that is never introduced anywhere (the expert reader of course still can infer what is likely meant) and (iii) are used in the algorithm to certify that a graph is a MAG without any discussion how induced paths are related to MAGs (indeed condition (i) in line 161 corresponds to no induced path existing between nonadjacent nodes, which is however not mentioned). Similar gaps exist in many places in the paper.

**Novelty:**
Separation routines that ensure the absence of almost directed paths are not novel and were also discussed in Chen et al. (2021). The routine for identifying inducing paths may be novel in the context of learning MAGs with an integer programming formulation, but detecting inducing paths has been generally discussed before in the literature, see for example (Wienöbst et al., A new constructive criterion for Markov equivalence of MAGs, Section 7) remarking that detecting inducing path takes time O(n^2 * m) for graphs with n variables and m edges. More generally, using quadratic programming for causal discovery with linear dependencies and a least-squares loss, while novel for MAGs, has been proposed in earlier work (Rytir et al. 2024) and this paper is an iterative extension of that framework.

**Simulations:**
The paper compares the proposed algorithm against other methods such as FCI using the SHD as metric. It is not mentioned how this is done fairly, with the FCI algorithm returning a PAG (a different graph type) and ExMAG returning a MAG. After looking at the code, it appears that for comparison the evaluation script only takes the directed and bidirected edges of the PAG into account, simply dropping all other edges. Hence, the simulation results need to be fundamentally questioned.

**Questions:**

**Questions**

1. Can you connect the score function (4) to the models proposed in equation (2) and (3)? Are there theoretical guarantees that the score function (asymptotically) identifies the MAG of the underlying causal structure?

2. In Figure 1 you show that ExMAG returns the graph G <-> A. Indeed, from a theoretical standpoint, both G <-> A and G -> A (as well as G <- A) are Markov equivalent and thus all explain the data equally well. Can you clarify why G <-> A is preferred by the algorithm here and what assumptions justify this?

3. SHD comparing the PAG returned by FCI with the MAG returned by ExMAG:
How do you do the comparison between the two graph types? The code below suggests that all but the directed or bidirected edges are dropped. Can you clarify whether that is the case?
```
    W_est = np.zeros((n, n))
    Wbi = np.zeros((n, n))
    for i in range(n):
        for j in range(n):
            if i == j:
                continue
            # i --> j
            if graph.graph[j,i] == 1 and graph.graph[i,j] == -1:
                W_est[i,j] = 1
            elif graph.graph[j,i] == 1 and graph.graph[i,j] == 1:
                Wbi[i,j] = 1

    return W_est, Wbi
```



**Suggestions:**

1. Please ensure a consistent citation style (quite often the parentheses are missing, such as "...causal learning Schölkopf & von Kügelgen (2022)" in line 023).

2. In line 393, it sounds like bow-free graphs are a subset of MAGs. All MAGs are bow-free.

3. Please check carefully for typos. Here are just a few I stumbled over:
- line 049: no space before "(Bühlmann...)"
- line 088: "Dissertation Hu (2023)"?
- line 113: "in *the* supplementary materials"
- line 131: "ancestorsof"
- line 150: "if *it* is contained" (also note here that non-overlapping path may not be a well-known term and that the "mathematical form" describes not a single collider but all with u as center node)
- line 194: "*Connection* to Causality"
- line 218: "is *the* lq norm"
- line 322: "become *a* MAG"

---

> ### Author Response · Authors · 2025-12-04
>
> The authors would like to thank the reviewers for their work and for helping us improve the manuscript. Main changes we did based on your comments include rewriting Section 3 (page 4), simplifying the notation, and correcting a problem with our interpretation of results returned by the FCI algorithms. Please see our specific comments below.
>
> > Directly in the abstract: "In mixed graphs, there are both directed and undirected  edges. An extensions [...] to this setting is known as maximally ancestral graphs. [...] the undirected edges represent confounding.", only later it is clarified that the undirected edges refer to bidirected edges
>
> Thank you, we have made sure to unify notation, and bidirected edges are called bidirected everywhere.
>
> > In the first paragraph "Dynamic Bayesian networks (DBN) are a popular model where a DAG represents causal relationships. The key challenge in learning DBNs is confounding.", with large parts of the paper then talking about DAGs and MAGs without the time series setting that DBNs address.
>
> Thank you for pointing this out. The reference to DBNs created unnecessary confusion.
> Our method is entirely focused on the static case; we therefore removed the DBN motivation and revised the introduction so it consistently discusses static SEMs and MAGs only.
>
> > On the second page "MAGs can represent feedback loops", which is simply a false statement.
>
> Thank you, we corrected the statement.
>
> > Can you connect the score function (4) to the models proposed in equation (2) and (3)? Are there theoretical guarantees that the score function (asymptotically) identifies the MAG of the underlying causal structure?
>
> The objective consists of two parts - the first one is the least squares error (assuming q=2) of the predictor, which is known to be optimal under the Gaussian noise assumption in linear regression problems. The second part is regularization that pushes down parameter values in the case of dependent or nearly-dependent features. We changed the original objective function and provided a reference to a paper where a similar objective can be found.
>
> > In Figure 1 you show that ExMAG returns the graph G <-> A. Indeed, from a theoretical standpoint, both G <-> A and G -> A (as well as G <- A) are Markov equivalent and thus all explain the data equally well. Can you clarify why G <-> A is preferred by the algorithm here and what assumptions justify this?
>
> We are aware that G->A and G<-A are indistinguishable from each other using only data with no other knowledge. Reformulating Your question, if y=5x + noise, then also x = 1/5 y + noise’.Therefore, if there is freedom in the DAG structure to include either G->A or A->G, one of them is used depending on the size of the error term in (4) and the regularization settings.
> The point of your question was, however, targeted more at the G <--> A setting. Here, the situation is similar; however, this time, we can use background knowledge stemming from the matrix F, which serves as an input matrix, denoting which pairs of variables which are not in direct cause-effect relationship. By definition (see, for example, presentation https://ccc.inaoep.mx/~esucar/Causalidad/MAGs_part1_fundamentals.pdf, slide 21, NOT OUR WORK), the G->A edges do not rule out the possibility of latent confounding, but G<->A assumes that neither G->A, nor A->G. Therefore, if the respective field in matrix F is marked true, G->A and A->G are both forbidden to the program, and G <-> A is used instead, in the case that it would explain the data well.
>
> > SHD comparing the PAG returned by FCI with the MAG returned by ExMAG: How do you do the comparison between the two graph types? The code below suggests that all but the directed or bidirected edges are dropped. Can you clarify whether that is the case?
>
> We fixed this issue. All edge types are now correctly extracted from the FCI solution. Accordingly, we updated the SHD and F1 score computations for PAGs. Please see the updated material and source code. We also re-ran the experiments. The FCI results are now slightly better, but ExMAG still performs better in most cases.
>
> > Suggestions:
> > Please ensure a consistent citation style (quite often the parentheses are missing, such as "...causal learning Schölkopf & von Kügelgen (2022)" in line 023).
>
> Thank you. We will make sure to be more careful with our citations in the future.
>
> > In line 393, it sounds like bow-free graphs are a subset of MAGs. All MAGs are bow-free.
>
> Thank you, we removed the formulation, as it was unnecessary with the formal definition following immediately afterwards.
>
> > Please check carefully for typos. Here are just a few I stumbled over:
>
> Thank you very much for pointing out the typos. We have fixed them and checked the paper once over.

---

### Official Review · Reviewer_yWWw · 2025-10-30

**Soundness:** 2
**Presentation:** 2
**Contribution:** 2
**Rating:** 2
**Confidence:** 3

**Summary:**

This paper investigates the issue of score-based causal discovery of maximal ancestral graphs (MAGs). The authors focus on a parametric family of causal models that consist of linear functions with non-linear components for unobserved confounders. Based on these parametric assumptions, they define a score function to evaluate how well a causal graph fits the observed data. Specifically, this score function uses the standard Lp-norm distance, incorporating additional regularization terms that penalize graph complexity. To ensure that the algorithm favors MAGs, the authors translate the inducing-path constraints of MAGs into integer constraints. As a result, the structural learning problem is transformed into a mixed integer quadratic program. Finally, simulations were conducted to validate the proposed algorithms.

**Strengths:**

The organization of this paper is generally clear. This paper studies an exciting problem in causal discovery for MAGs with unobserved confounders. Popular score-based methods focus on Markovian causal graphs without unobserved confounders. Additionally, the inequality constraints enforcing the inducing paths are novel.

**Weaknesses:**

- The choice of the Lp-norm as the score function is intriguing. Typically, the design of the score function derives from approximating the free Bayes energy. For Markovian causal models without unobserved confounders (UCs), the free Bayes energy can be approximated using the standard Bayesian Information Criterion (BIC). However, in causal models that include UCs, issues of singularities can emerge, making BIC an inadequate score for graph selection. Watanabe (2013) provided a solid introduction to this problem, introducing a generalized BIC score known as WBIC, which can be applied to singular models. It is important to note that the Lp-norm is generally not the optimal choice for causal discovery, as it can yield inconsistent results. Specifically, two Minimal Aggregated Graphs (MAGs) that have the same free Bayes energy may have different Lp-norm scores, and a MAG with optimal free Bayes energy does not necessarily correspond to a minimal Lp-norm score.

- Moreover, the formulation of mixed integer programs is typically computationally complex to solve. From this perspective, it remains unclear whether the current formulation offers any significant improvements over existing structural learning algorithms. For instance, in the reported simulations, the enhancements over the Fast Causal Inference (FCI) algorithm, a standard constraint-based approach, are not entirely consistent. The observed improvements may also be attributed to the fact that the proposed algorithm relies on additional parametric assumptions.

**Questions:**

- Is the proposed score function consistent for causal discovery with unobserved confounders?
- Is it possible to approximate the resulting mixed integer quadratic program using a differentiable optimization problem?

---

> ### Author Response · Authors · 2025-12-04
>
> The authors would like to thank the reviewers for their work and for helping us improve the manuscript. Main changes we did based on your comments include rewriting Section 3 (page 4), simplifying the notation, and correcting a problem with our interpretation of results returned by the FCI algorithms. Please see our specific comments below.
>
> > The choice of the Lp-norm as the score function is intriguing. Typically, the design of the score function derives from approximating the free Bayes energy. For Markovian causal models without unobserved confounders (UCs), the free Bayes energy can be approximated using the standard Bayesian Information Criterion (BIC). However, in causal models that include UCs, issues of singularities can emerge, making BIC an inadequate score for graph selection. Watanabe (2013) provided a solid introduction to this problem, introducing a generalized BIC score known as WBIC, which can be applied to singular models. It is important to note that the Lp-norm is generally not the optimal choice for causal discovery, as it can yield inconsistent results. Specifically, two Minimal Aggregated Graphs (MAGs) that have the same free Bayes energy may have different Lp-norm scores, and a MAG with optimal free Bayes energy does not necessarily correspond to a minimal Lp-norm score.
>
> Thank you for your comment. Our goal is thus not to approximate WBIC or derive a Bayesian score for MAG learning. Instead, we present an alternative optimization-based approach using a regression-based formulation derived from a linear-Gaussian SEM. The objective consists of two parts - the first one is the least squares error (assuming q=2) of the predictor, which is known to be optimal under the Gaussian noise assumption in linear regression problems. The second part is regularization that pushes down parameter values in the case of dependent or nearly-dependent features.
>
> > Moreover, the formulation of mixed integer programs is typically computationally complex to solve. From this perspective, it remains unclear whether the current formulation offers any significant improvements over existing structural learning algorithms. For instance, in the reported simulations, the enhancements over the Fast Causal Inference (FCI) algorithm, a standard constraint-based approach, are not entirely consistent. The observed improvements may also be attributed to the fact that the proposed algorithm relies on additional parametric assumptions.
>
> As one of the other reviewers pointed out to us, we made an error in interpreting FCI results, which resulted in ignoring some of the edges returned by FCI. We fixed this issue. All edge types are now correctly extracted from the FCI solution. Accordingly, we updated the SHD and F1 score computations for PAGs. Please see the updated material and source code. We also re-ran the experiments. The FCI results are now slightly better, but ExMAG still performs better in most cases.
>
> > Is the proposed score function consistent for causal discovery with unobserved confounders?
>
> While the objective is similar in both cases, it is not exactly the same - when we know the cofounder value, it can have different weights for both the edges going from the cofounder into the effects. Also, the regularization is different, as the number of edges in the graph is different.
>
> > Is it possible to approximate the resulting mixed integer quadratic program using a differentiable optimization problem?
>
> While the least squares error is differentiable, and some of the acyclicity constraints can be formulated as differentiable optimization problems (for example, NOTEARS, which computes the trace of the matrix-exponential of the adjacency matrix, which is zero only for DAGs), enforcing MAG properties requires global combinatorial constraints, for which we are not aware of any obvious smooth relaxations.

---

### Official Review · Reviewer_u9dP · 2025-10-31

**Soundness:** 1
**Presentation:** 2
**Contribution:** 1
**Rating:** 2
**Confidence:** 5

**Summary:**

In their work, the authors focus on learning causal models from data. The main result is a proposed algorithm called ExMAG, which learns Maximal Ancestral Graph (MAG) from data. The algorithm is based on the formulation of the structure learning task via a mixed integer quadratic program (MIQP). The paper presents comparative experimental results for the algorithm.

**Strengths:**

Learning causal structure from observed data is an important and very challenging task in causality and ML in general. In particular, the problem becomes exceptionally difficult (and remains open) if we assume both latent confounders and selection bias, as is the case in the MAG model. So the problem studied in the work is important and extremely timely in the field.

**Weaknesses:**

The paper presents an ad hoc heuristic whose properties have not been analyzed by the authors. The comparative experimental results are not convincing because the authors use incorrect measures of the quality of the algorithm results (or the given description is incomplete). Furthermore, the paper contains a number of incorrect formulations, erroneous statements and ambiguities. Details below.

The authors claim to have developed an algorithm for learning MAGs. However, in their paper, they only consider latent confounding and completely ignore selection bias.

The claim: "We propose a score-based branch-and-cut algorithm for learning maximally ancestral graphs" is wrong. The presented approach is not score-based.

There are several ambiguities in the article regarding (latent) confounders. In Figure 1 the authors claim that ExMAG correctly identifies the confounding in (c). But in the Berkeley graduate admission example (a) there is no confounder between Gender and Admit at all. Moreover, to explain the example in Figure 1, it is necessary to write which variables are latent and what D_A and D_B mean.

There is no precise definition for the ground truth data (linear) model assumed in the work. The equation in L. 178 is not clear enough. For example, what is the meaning of g, \tilde{w}, w_{0,j} in eq. (1)? What is the relationship between Y and X? In MILP formulation in Section 3.2 Y is completely ignored.

There is no analysis of the correctness of using  Eq.(4) in causal structure learning.

The authors use incorrect measures of the quality of the algorithm results (or the description is incomplete): As follows from the description in Section 5 the authors  measure SHD between MAGs. But it is wrong. Instead, one should compare partial ancestral graphs (PAGs), which characterize the class of Markov equivalent solutions, with PAGs for ground-truth.

It is not clear why the running times of FCI in experiments presented in Section 5 are 0 (or very close to 0).

The algorithms presented in Section 4 are fairly standard and well known in the graphical causality community.

The claim that "Rantanen et al. (2021) explored a constraint-based approach for MAG discovery, leveraging conditional independence testing" is false. In fact, Rantanen et al.  develop methodology for *score-based* structure learning of directed maximal
ancestral graphs. They do not use conditional independence tests at all.

**Questions:**

Please refer to my comments above. Moreover, what is the point of using *not* bow-free (BF) graphs in experiments if the definition of MAG assumed in the work satisfies the BF property?

---

> ### Author Response · Authors · 2025-12-04
>
> The authors would like to thank the reviewers for their work and for helping us improve the manuscript. Main changes we did based on your comments include rewriting Section 3 (page 4), simplifying the notation, and correcting a problem with our interpretation of results returned by the FCI algorithms. Please see our specific comments below.
>
> > The authors claim to have developed an algorithm for learning MAGs. However, in their paper, they only consider latent confounding and completely ignore selection bias.
>
> Thank you, we clarified the introduction.
>
> > The claim: "We propose a score-based branch-and-cut algorithm for learning maximally ancestral graphs" is wrong. The presented approach is not score-based.
>
> Thank you. Paper (not our work):
> Scutari, Marco, et al. "Who learns better bayesian network structures: Constraint-based, score-based or hybrid algorithms?." Int. Conf. on Probabilistic Graphical Models. 2018.
> defines score-based algorithms as: “Score-based algorithms represent the application of general optimisation techniques to BN structure learning. Each candidate DAG is assigned a network score reflecting its goodness of fit, which the algorithm then attempts to maximise.”
> While we do not explicitly formulate the score, we believe that our algorithm matches this definition with the least squares error as the score, similarly to BIC. To eliminate any possible confusion, we removed this phrase from the paper.
>
> > There are several ambiguities in the article regarding (latent) confounders. In Figure 1 the authors claim that ExMAG correctly identifies the confounding in (c). But in the Berkeley graduate admission example (a) there is no confounder between Gender and Admit at all. Moreover, to explain the example in Figure 1, it is necessary to write which variables are latent and what D_A and D_B mean.
>
> Thank you for your comment. We will simplify the example. The department is the hidden variable.
>
> > There is no precise definition for the ground truth data (linear) model assumed in the work. The equation in L. 178 is not clear enough. For example, what is the meaning of g, \tilde{w}, w_{0,j} in eq. (1)? What is the relationship between Y and X? In MILP formulation in Section 3.2 Y is completely ignored.
>
> In our text, we motivated the method by the dynamic Bayesian networks; however, the paper focuses mostly on the static case, which brought some confusion. X serves as an observed variable at time zero, Y values are observed in the next time steps. In the static case, the X variable represents the input data, and also works as the data predicted by the model (i.e., X=Y). To improve readability, we decided to completely redo this part of the text.
>
> > There is no analysis of the correctness of using Eq.(4) in causal structure learning.
>
> The objective consists of two parts - the first one is the least squares error (assuming q=2) of the predictor. For linear structural equations with Gaussian noise, the negative-log likelihood decomposes to least squares error. Therefore, the term in our objective is equivalent to maximizing the likelihood of a linear-Gaussian SEM over the observed variables.
> We changed the original objective function and provided a reference to a paper where a similar objective can be found.
>
> > The authors use incorrect measures of the quality of the algorithm results ...: the authors measure SHD between MAGs. But it is wrong. Instead, one should compare PAGs, which characterize the class of Markov equivalent solutions, with PAGs for ground-truth.
>
> Our synthetic experiments provide a single ground-truth MAG, not a class of equivalent MAGs. The PAGs contain extra information - wildcards representing that the direction of the edges can be arbitrary. Nevertheless, for our ground truth data, we do not have the PAG available.
>
> > It is not clear why the running times of FCI in experiments presented in Section 5 are 0 ...
>
> The FCI algorithm runs fast - even on the largest instances, it finishes under one minute. In the plots, this looks like 0.
>
> > The claim that "Rantanen et al. (2021) explored a constraint-based approach for MAG discovery, leveraging conditional independence testing" is false. In fact, Rantanen et al. develop methodology for score-based structure learning of directed maximal ancestral graphs. ...
>
> Thank you, we corrected the text to “The same year, \cite{Rantanen2021exactsearch} explored a score-based approach for directed MAG discovery, leveraging a local score function optimized using pruning rules and dynamic programming.”.
>
> > Moreover, what is the point of using not bow-free (BF) graphs in experiments if the definition of MAG assumed in the work satisfies the BF property?
>
> Although MAGs are bow-free by definition, we include some non-BF graphs in experiments to evaluate the robustness of ExMAG when run on graphs that do not fulfill all assumptions of the algorithm.

---

### Official Review · Reviewer_LKKY · 2025-11-01

**Soundness:** 2
**Presentation:** 3
**Contribution:** 2
**Rating:** 4
**Confidence:** 3

**Summary:**

This work introduces ExMAG, a score-based branch-and-cut method for learning Maximally Ancestral Graphs (MAGs) that represent latent confounding via bidirected edges. The approach formulates structure learning as a MIQP and uses lazy constraints to enforce acyclicity, ancestral consistency, and maximality. Experiments on synthetic and financial data report competitive SHD and runtime, with scalability demonstrated up to 25 variables.

**Strengths:**

1. **Clear exposition of the optimization framework.**

The formulation of the MIQP problem is well explained, and the authors explicitly relate each constraint type to known graph-theoretic properties (acyclicity, ancestry, maximality).

2. **Empirical validation.**

 Experiments are diverse and compare against key baselines — FCI, ip4ancadmg, and ipbm. Results convincingly show improved SHD and runtime, with practical scalability benefits.

**Weaknesses:**

1. **Lack of theoretical completeness.**

While the approach draws on guarantees from prior MIP formulations (Chen, 2021; Dash, 2025), the paper does not establish new proofs that ExMAG preserves acyclicity and ancestral correctness under its specific implementation. Maximality — a defining property of MAGs — is enforced only via iterative inducing-path checks, without a completeness or convergence guarantee. As a result, it remains unclear whether the final output is always a valid MAG.

2. **Potential loss of global optimality.**

The reliance on lazy constraint separation preserves optimality only when all relevant constraints can be identified. Because the inducing-path separation routine is heuristic, ExMAG may return a solution that is optimal under an incomplete constraint set. The paper does not theoretically analyze or empirically quantify this failure mode.

3. **Limited causal interpretation.**

The objective optimizes a regression-based reconstruction loss with sparsity regularization, not a likelihood or identifiable SEM formulation. Unlike approaches grounded in explicit causal assumptions (e.g., NOTEARS), the causal semantics of the recovered edges remains ambiguous.

**Questions:**

* Could the authors formalize or at least characterize sufficient conditions under which the inducing-path separation routine is guaranteed to be complete? Is there a finite or provably convergent set of constraints that ensures the resulting graph is a true MAG?

* Could the authors derive the objective from a Structural Equation Model (SEM) assumption (e.g., linear Gaussian noise), or clarify under what conditions the quadratic loss approximates a log-likelihood?

---

> ### Author Response · Authors · 2025-12-04
>
> The authors would like to thank the reviewers for their work and for helping us improve the manuscript. Main changes we did based on your comments include rewriting Section 3 (page 4), simplifying the notation, and correcting a problem with our interpretation of results returned by the FCI algorithms. Please see our specific comments below.
>
> > Lack of theoretical completeness.
> > While the approach draws on guarantees from prior MIP formulations (Chen, 2021; Dash, 2025), the paper does not establish new proofs that ExMAG preserves acyclicity and ancestral correctness under its specific implementation. Maximality — a defining property of MAGs — is enforced only via iterative inducing-path checks, without a completeness or convergence guarantee. As a result, it remains unclear whether the final output is always a valid MAG.
>
> If all constraints forbidding possible inducing paths, directed cycles, and almost directed cycles were included in the program, the program would clearly find the MAG optimizing equation (4). This is, however, infeasible, as there are exponentially many such constraints.
>
> Instead, we start with a relaxed program that contains none of them. Its optimum is necessarily at least as good as the solution to the original problem, and therefore provides a lower bound on the optimum of the full program. In each iteration, either we reach a solution in which no inducing paths (dc/adc) are present—implying that we have found an optimum—or we forbid at least one of the violated constraints, thereby moving toward the complete program. The relaxed program remains smaller, as it avoids the potentially exponential number of constraints, and we add only those that are actually violated, one by one or in batches. Since there is only a finite number of such constraints and each iteration eliminates at least one violation, convergence is guaranteed. Once we arrive at a program in which no inducing paths and no (almost) directed cycles exist, we know that the solution is optimal, as it is a relaxation of the original program (omitting only inactive constraints) and thus constitutes a valid lower bound on the optimum of the full problem.
> This strategy is a common technique used in Branch-and-Bound approaches for problems such as solving integer linear programs.
>
> > Potential loss of global optimality.
> > The reliance on lazy constraint separation preserves optimality only when all relevant constraints can be identified. Because the inducing-path separation routine is heuristic, ExMAG may return a solution that is optimal under an incomplete constraint set. The paper does not theoretically analyze or empirically quantify this failure mode.
>
> The inducing path is defined as a path where every inner vertex is a collider (in other words, both edges of the path point into it), and it is an ancestor of one of the endpoints of the path. This means that all inducing paths can be found by doing DFS on all vertices on the graph of bidirected edges. This set is pruned using the ancestor relationship found by the Floyd-Warshall algorithm on the directed edges graph, so that only inducing paths are found. As a result, whenever there exists an inducing path, it is found.
>
> > Could the authors formalize or at least characterize sufficient conditions under which the inducing-path separation routine is guaranteed to be complete? Is there a finite or provably convergent set of constraints that ensures the resulting graph is a true MAG?
>
> Please, see the answer above.
>
> > Could the authors derive the objective from a Structural Equation Model (SEM) assumption (e.g., linear Gaussian noise), or clarify under what conditions the quadratic loss approximates a log-likelihood?
>
> The objective consists of two parts - the first one is the least squares error (assuming q=2) of the predictor. For linear structural equations with Gaussian noise, the negative-log likelihood decomposes to least squares error. Therefore, the term in our objective is equivalent to maximizing the likelihood of a linear-Gaussian SEM over the observed variables. Nevertheless, we provide the option to set q=1, as an approximation, which scales further.
> The second part is regularization that pushes down parameter values in the case of dependent or nearly-dependent features.
> We slightly changed the original objective function and provided a reference to a paper where a similar objective can be found.

---

### Note · Program_Chairs · 2026-01-17
**Submission Desk Rejected by Program Chairs**

The following references in this submission do not refer to real documents and/or have major errors in bibliographic information:

 A. P. Dawid and V. Didelez. Causal inference in graphical models. Journal of Causal Inference, 2 (1):22-38, 2010. doi: 10.1214/10-JCI260.
M. Rojas-Carulla et al. Causal inference and distributional robustness. Statistical Science, 33(3): 432-445, 2018. doi: 10.1214/18-STS635.